# Non-Coding RNAs in Regulating Plaque Progression and Remodeling of Extracellular Matrix in Atherosclerosis

**DOI:** 10.3390/ijms232213731

**Published:** 2022-11-08

**Authors:** Drishtant Singh, Vikrant Rai, Devendra K. Agrawal

**Affiliations:** Department of Translational Research, College of Osteopathic Medicine of the Pacific, Western University of Health Sciences, Pomona, CA 91766, USA

**Keywords:** atherosclerosis, epigenetics, extracellular matrix remodeling, circRNAs, lncRNAs, miRNAs

## Abstract

Non-coding RNAs (ncRNAs) regulate cell proliferation, migration, differentiation, inflammation, metabolism of clinically important biomolecules, and other cellular processes. They do not encode proteins but are involved in the regulatory network of various proteins that are directly related to the pathogenesis of diseases. Little is known about the ncRNA-associated mechanisms of atherosclerosis and related cardiovascular disorders. Remodeling of the extracellular matrix (ECM) is critical in the pathogenesis of atherosclerosis and related disorders; however, its regulatory proteins are the potential subjects to explore with special emphasis on epigenetic regulatory components. The activity of regulatory proteins involved in ECM remodeling is regulated by various ncRNA molecules, as evident from recent research. Thus, it is important to critically evaluate the existing literature to enhance the understanding of nc-RNAs-regulated molecular mechanisms regulating ECM components, remodeling, and progression of atherosclerosis. This is crucial since deregulated ECM remodeling contributes to atherosclerosis. Thus, an in-depth understanding of ncRNA-associated ECM remodeling may identify novel targets for the treatment of atherosclerosis and other cardiovascular diseases.

## 1. Introduction

Cardiovascular disease has been a major cause of death worldwide, posing serious concerns to human health for the past 20 years [1]. The pathophysiological basis for many prevalent cardiovascular disorders is atherosclerosis, a chronic inflammatory disease [2]. Lipoproteins deposit in the subintimal region, and subsequent oxidative responses mediate the process of plaque formation and the progression of atherosclerosis, which is more common in big or medium-sized arteries [3]. This is accompanied by macrophage recruitment and foam cell formation, movement of vascular smooth muscle cells (VSMCs) to the intima, and progression of atherosclerotic plaque. Plaques that extend into the artery cause duct stenosis, and rupture of plaque gives rise to emboli causing adverse ischemic events [4]. Remodeling of the extracellular matrix (ECM) is a major process involved in atherogenesis that changes the vasculature and affects its regulation. The early change in the atherosclerosis starts with deposition of fibronectin which is then predominantly occupied by deposition of collagen and cross-linking [5]. Subsequently, various complex changes like degradation of ECM proteins begin that lead to rupture of the plaque [6]. Although the pathophysiology of atherosclerosis and other cardiovascular diseases have been extensively studied, epigenetic regulation is largely not well studied. Therefore, it is critical to explore the epigenetic regulation of various molecular mechanisms involved in the pathogenesis of atherosclerosis to identify novel therapeutic strategies.

The discovery of non-coding ribonucleic acids (ncRNAs) changed our understanding of the post-translational, post-transcriptional, and epigenetic regulation of gene expression in controlling cellular homeostasis in various diseases. Recent breakthroughs in the field of genomics, facilitated by technologies such as chromatin immune-precipitation RNA sequencing (ChIP RNA Seq), Assay for Transposase-Accessible Chromatin (ATAC) seq, transcriptome analysis, and next-generation sequencing (NGS), have provided fresh insights and fundamentally altered our knowledge of small ncRNA molecules, which were long regarded as “junk DNA.” The fact that about 99% of the genome consists of non-coding DNA and approximately 1% codes for functional proteins demonstrates the intricacy and significance of ncRNAs in regulating gene expression [7,8,9]. Regulatory ncRNAs such as microRNAs (miRNAs; miRs) and long non-coding RNAs (lncRNAs) have had a profound impact on research in numerous domains, like cancer [10,11,12], cardiovascular diseases [13,14,15], and diabetes [16,17,18]. The epigenetic regulation of these ncRNAs is crucial in both the early development and the etiology of heart diseases [19,20,21].

Emerging approaches based on genomic data have changed diagnostic and therapeutic procedures, allowing for the early detection of problems, and providing hope for more successful treatments. The purpose of this article is to offer an up-to-date account of the involvement of noncoding RNAs (ncRNAs) in cardiovascular diseases with an emphasis on the regulation of extracellular matrix remodeling in atherosclerosis.

## 2. Extracellular Matrix

The extracellular matrix comprises various cells and cellular structures that constitute atherosclerotic plaque scaffolding. It is made up of different structural components that are regulated by a class of different regulatory factors. Collagens, hyaluronan, elastic fibers, proteoglycans, and many glycoproteins are essential elements of vascular ECMs that are all coupled in a complex dynamic 3D matrix system. This link controls the biomechanical properties of arteries and the phenotype of the cells such as ECs, VSMCs, adventitial fibroblasts, and immune cells invading circulation. VSMCs are the predominant cell types identified in terms of their ability to produce ECM macromolecules [22]. The development of atherosclerosis commences with focal endothelial cell injury in arteries, which enhances the invasion of freely circulating monocytes and T lymphocytes [23]. Monocytes differentiate into macrophages in the subendothelial intima, where they release cytokines and aggravate the inflammatory environment and endocytose LDL fragments and then become lipid-laden foam cells. Concurrently, the SMCs from the medial layer migrate into the intima, proliferate, and make collagen fibers to form a fibrous cap that stabilizes the intima. In contrast, forming a lipid-rich malignant core destabilizes the lesion, eventually leading to erosion in high-risk, rupture-prone plaques, causing thrombosis, which could develop arterial obstruction, resulting in myocardial infarction (MI) [24,25].

### ECM Remodelling and Atherosclerosis

Throughout life, the structure and function of the vasculature are determined by the interaction between various ECM components. In the early phase of atherosclerosis, proteoglycans comprise the majority of the ECM, but as the disease progresses, collagens become the predominant ECM component, accompanied by a decrease in elastic fibers, glycoproteins, and proteoglycans [26]. Moreover, elastin and collagen are the most thoroughly researched ECM proteins in the etiology of atherosclerosis. Uncontrolled degradation of these proteins increases the course of atherosclerosis because it permits transendothelial migration of leukocytes, VSMC migration and proliferation, neovascularization, vascular cell death, neointima formation, and ultimately the rupture of the arterial wall [27]. Elastic fibers, composed of elastin components, stabilize collagen; hence, damaged elastin (seen in plaques), resulting in the inability to generate a stable matrix, which leads to plaque rupture [28].

Lipoproteins containing apo-B penetrate the arterial intimal endothelial layer and concentrate in the subendothelial region, where they are endocytosed by intimal macrophages. Simultaneously, local blood flow disruptions (non-linear flow) in atherosclerosis-prone locations (e.g., arterial branches) result in decreased shear stress, which is recognized by endothelial cells during the mechano-transduction process. These processes alter the microenvironment of the arterial wall intima, promoting subsequent changes such as foam cell generation, VSMC migration and conversion from contractile to synthetic phenotypes, ECM matrix remodeling, and necrotic core formation and calcifications [25,29]. Furthermore, the immune system is critical in the pathogenesis of atherogenesis [30]. Many plaques mature into stable structures that cause the chronic coronary syndrome, while some of them undergo ultrastructural changes that make them prone to rupture. These plaques are referred to as ‘unstable’ or ‘susceptible’ [31]. According to the reported characteristics, susceptible plaque is a thin cap fibroatheroma (TCFA) with a necrotic core and an overlaying fibrous cap of 65µm thickness [32]. The presence of susceptible plaques is required for the occurrence of significant cardiovascular adverse effects [33]. As a result, medications aiming to stabilize atherosclerotic plaque are required. However, due to clinical denial, ideal therapy focused on stabilizing plaques must focus on the advancement of molecular stabilizing paths ‘in general’ [34,35,36,37,38,39,40] instead of on the stabilization of specific atherosclerotic lesions.

## 3. Non-Coding RNA

Transcripts that are not translated into polypeptides or proteins are known as non-coding RNAs (ncRNAs). Approximately 1–2% of genes are responsible for making proteins, suggesting that there are many non-coding genes with unidentified roles [41]. These ncRNAs demonstrate various biological roles and directly participate in several physiological processes [42]. Prior research on ncRNAs focused mostly on their regulatory functions within cells. Later, extracellular vesicles (EVs) were thoroughly investigated, and it was discovered that these EVs contain lipids, proteins, messenger RNAs (mRNAs), and ncRNAs with biological functions [43,44]. The ncRNAs typically exist in EVs or attach to proteins or lipids to avoid ribonuclease-mediated destruction [45]. The majority of ncRNAs in the blood are either contained in EVs [46] or are protein-bound, such as with lipoproteins [47], argonaute protein (AGO2), and nucleophosphoprotein 1 (NPM1) [48]. Recent research revealed that extracellular ncRNAs control intracellular gene expression, mediate intercellular communication, and are intimately connected to numerous pathogenic processes [49,50,51,52,53]. Figure 1 represents the various cellular processes being controlled by the action of various ncRNAs.

According to recent findings, extracellular ncRNAs are thought to be closely associated with atherosclerosis. Extracellular ncRNAs are crucial regulators of many cells, including immune cells, macrophages, and endothelial cells. They play a role in atherosclerotic processes such as angiogenesis, foam cell formation, and atherosclerotic plaque progression and rupture [41,54,55]. Extracellular ncRNAs serve as a useful diagnostic indicator and potential treatment target in atherosclerosis [56]. The association of ncRNAs to disease development, diagnosis, therapy, and incidence, in the past few years has seen a major advancement in the study of ncRNAs related to cardiovascular disorders. This review article critically discusses the specific function, putative mechanism, and prospective applications of ncRNAs in atherosclerosis, emphasizing miRNAs, circRNAs, and lncRNAs-mediated ECM remodeling in atherosclerosis.

### 3.1. miRNAs and Atherosclerosis

The miRNAs are genetically conserved, containing 18–24 nucleotides, small single-stranded non-coding RNAs that regulate gene expression at the post-transcriptional level by binding to the 3′-untranslated region of certain target mRNA sequences, thereby reducing protein synthesis by inhibiting mRNA translation [10,57,58,59]. There are more than 60% of human protein-coding genes have miRNA target sites in their 3′-UTR, and various studies have shown the involvement of miRNA/mRNA interactions as the key regulatory network in different biological processes [60,61,62]. With the unique characteristics of miRNAs, these have been extensively used as key regulators of mRNA and protein expression in many diseases, including cardiovascular diseases [61,62,63,64]. Various studies (Table 1) have analyzed the role of miRNAs in atherosclerosis and ECM remodeling.

The expression of MiR-1a-3p, miR-1b-5p, and miR-1 was found to be the most prominently increased in different diseases related to subclinical atherosclerosis. The miR-1 mimics can activate endothelial inflammation through increased production of E-selectin, intercellular adhesion molecule (ICAM)-1, and vascular cell adhesion molecule (VCAM)-1 at both the mRNA and protein levels. The in-vivo findings showed that miR-1 knockdown by antagomiR-1 reversed the endothelial and inflammatory activation at the lesion site, revealing a novel therapeutic target for atherosclerosis [19]. Wu et al. [111] demonstrated that miR-142-5p targeted myocardin-like protein 2 to drive the amplification and migration of human aortic smooth muscle cells, promoting atherosclerosis. Su et al. [112] reported the presence of miR-181a-5p and miR-181a-3p in atherosclerotic lesions of ApoE mice fed with a high-fat diet and in the plasma of patients with coronary artery disease (CAD). These findings indicate the potential role of these two miRNAs in atherogenesis. Also, the overexpression of miR-181a-5p and miR-181a-3p in ApoE mice decreased the plaque size. In contrast, the gain-of-function mutation decreased inflammatory genes like ICAM-1 and VCAM-1 and leukocyte infiltration in the aortic intima.

Raitoharju et al. [113] reported 58 miRNAs that were differentially expressed between atherosclerotic plaques and non-atherosclerotic left internal thoracic arteries. Of these, up-regulated five miRs viz. miR-21, miR-34a, miR-146a, miR-146b-5p, and miR-2010 were involved in the regulation of 187 mRNAs in atherosclerotic plaques. The proteins translated from these genes were involved in signal transduction, transcription control, and vesicular transport. In another study, it was reported that miR-130a expression was increased in atherosclerotic mice. In an in-vitro model of atherosclerosis, miR-130a overexpression enhanced inflammatory factors like tumor necrosis factor (TNF)-α and interleukin (IL)-1, IL-6, and IL-8 and its downregulation reduced the inflammation by attenuating TNF-α, IL-1, IL-6, and IL-8. Furthermore, in the in-vitro model, over-expression of miR-130a might reduce peroxisome proliferator-activated receptor (PPAR) protein expression while inducing NF-κB protein expression. Still, its suppression promoted PPAR protein expression while suppressing NF-κB protein expression. PPAR activation inhibited the pro-inflammatory effects of miR-130a in an atherosclerosis-induced in-vitro model [114]. Polyakova et al. [115] reported that the SYNTAX (tool to score complexity of CAD) score I index and serum miR-203 expression level exhibited a positive association in patients with CAD. In the atrial myocardium of patients with triple vessel disease, miR-27a, miR-133a, and miR-203 expressions were substantially greater than those of patients with 1–2 vessel disease. This association was also observed for miR-27a, miR-133a, and miR-203 expressions in the blood. Another study revealed that targeting miR-33 in atherosclerotic macrophages by anti-miR-33 conjugated pH low-insertion peptide (pHLIP) constructs to inhibit miR-33 improves collagen formation and reduces lipid accumulation, thereby improving atherosclerotic regression. Additionally, a single-cell RNA sequencing study showed that macrophages from atherosclerotic lesions targeted by pHLIP-anti-miR-33 had lower levels of matrix metalloproteinase (MMP)-12 and greater levels of fibrotic genes (Col2a1, Col3a1, Col1a2, Fn1, etc.) and tissue inhibitor of metalloproteinase (TIMP)-3 [116].

Another study reported that the expression of three miRs, miR-129-1-3p, miR-4312, and miR-5196-3p differed significantly between the acute ischemic stroke and atherosclerosis/healthy control groups. Twelve pathways were affected by the miR-129-1-3p target genes, three of which were related to axonal and synaptic function: sphingolipid signaling, retrograde endocannabinoid signaling, and axon guidance. Cortical neurite length and Runx2 levels were considerably reduced by miR-129-1-3p mimics, whereas Runx2 expression was elevated, and neurite growth was boosted by miR-129-1-3p inhibitors [117].

Egea et al. [118] demonstrated that the treatment of human mesenchymal stem cells (hMSCs) with LL-37 boosted let-7f and N-formyl peptide receptor 2 (FPR2) production, which ultimately helped in the stabilization of atherosclerotic plaque. Circulating hMSCs attach to athero-prone endothelium more frequently in an ApoE animal model of atherosclerosis. High levels of let-7f in the hMSCs, as determined by two-photon laser scanning imaging and ex-vivo artery perfusion, contributed to increased attachment of MSCs. Additionally, the exposure of hMSCs to homogenized human atheromatous plaque material significantly increased the production of different cytokines, chemokines, MMPs, and TIMPs. Moreover, the exposure of hMSCs to human plaque extracts causes hMSCs to differentiate into cells of the myogenic lineage, indicating a potential stabilizing influence on the plaque.

### 3.2. circRNAs and Atherosclerosis

A group of RNA molecules known as circRNAs is produced through exon reverse splicing or intron lariat. Due to their closed ring structure, which shields them from the effects of RNA exonuclease, circRNA production is generally stable and tissue- and developmental stage-specific [119]. Due to the self-regulating, transposing, and other salient features of circRNAs, many studies have been recently conducted to investigate the role of circRNAs in the initiation and progression of atherosclerotic plaque and other cardiovascular diseases [120,121]. The studies have proposed the diagnostic value of different circRNAs in preventing and treating atherosclerosis (Table 2).

To discover circRNAs involved in atherosclerosis, human umbilical vein endothelial cells (HUVECs) stimulated with oxidized low-density lipoprotein (ox-LDL) were subjected to circRNA microarray analysis, where Hsa circ 0003575 showed the highest upregulation among all the circRNAs. Loss-of-function tests demonstrated that Hsa circ 0003575 inhibits endothelial cell (EC) growth, promotes apoptosis, and may act as a sponge for miRs miR-199-3p, miR-9-5p, miR-377-3p, and miR-141-3 [140]. Some circRNAs, such as ANRIL and LincP21, have significantly higher circulating levels and are associated with the severity of atherosclerosis [141,142].

CircRNAs play a major role in atherosclerosis and CAD [143,144,145,146]. In patients with CAD, nine circRNAs were reported by Pan et al. [147]. Ox-LDL treatment of HUVECs and feeding a high-fat diet to mice resulted in a downregulation of circHIPK3 expression, whereas overexpressing circHIPK3 increased autophagy, which was inhibited in atherosclerosis [143]. The expression of circHIPK3 was downregulated in mice on a high-fat diet and in ox-LDL-treated HUVECs. The level of autophagy was decreased in atherosclerosis, which was reversed by the overexpression of circHIPK3. Meanwhile, forced expression of circHIPK3 would reduce the accumulation of lipids in HUVECs.

In an atherosclerotic rabbit model, analysis of a variably expressed circRNA-miR-mRNA triple network showed that competition among circRNAs and their mRNAs might be a key factor in the onset of atherosclerosis [16]. When Hsa circ 0030042 was overexpressed, it acted as an internal eukaryotic initiation factor, inhibiting ox-LDL-induced aberrant autophagy in HUVECs, and sustaining plaque stability in-vivo. Furthermore, Hsa circ 0030042 inhibited autophagy by sponging eIF4A3 and preventing its recruitment to the mRNAs for beclin1 and forkhead box O1 (FOXO1), though the suppression of beclin1 and FOXO1 caused by Hsa circ 0030042 was offset by increased eIF4A3 expression or decreased Hsa circ 0030042 interaction. In ApoE^−/−^ rats fed a high-fat diet, Hsa circ 0030042 also increased plaque stabilization and reversed eIF4A3-induced plaque instability [148]. A microarray examining the circRNAs in the peripheral blood of CAD patients showed a strong correlation of hsa-circRNA11783-2 with the condition, and Hsa circ 0008507, Hsa circ 0001946, and Hsa circ 0000284 are independent risk factors for CAD [149]. Wang et al. [150] revealed that in CAD patients, 624 circRNAs and 171 circRNAs were significantly elevated and downregulated, respectively, compared to controls. In large cohorts, Hsa circ 0001879 and Hsa circ 0004104 were shown to be considerably elevated. The combination of Hsa circ 0001879 and Hsa circ 0004104, along with CAD risk variables, performed best in distinguishing CAD patients from healthy controls. Additionally, two non-coding RNA, namely, ANRIL (antisense non-coding RNA at the INK4 locus) and circANRIL (circular ANRIL), transcribed at the chromosome 9p21 region, were found to be associated with a high risk of cardiovascular disease. However, it was discovered that they had opposing effects on the onset of CAD. Although upregulation of circANRIL prevented the onset of CAD [151], upregulation of ANRIL was linked to an increase in the incidence of atherosclerosis [141].

The circRNAs are crucial in controlling the stability of atherosclerotic plaques, as in acutely ruptured carotid plaques. It was discovered that circRNA-16 was elevated while miR-221, which is linked to VSMC proliferation and death, was downregulated [152]. Axis inhibition protein 2 was another target of miR-221-3p, which enhanced the proliferation of pulmonary arterial smooth muscle cells. Therefore, through the miR-221/Ets-1 and AXIN2 axes, circRNA-16 may play a significant regulatory function in the stability of arterial plaques [153].

### 3.3. lncRNA and Atherosclerosis

Several lncRNAs with a role in atherosclerosis have been identified. lncRNAs are expressed in different cell types, present in atherosclerotic lesions, and have been implicated in several atherogenic processes, such as endothelial dysfunction and lipid deposition [154]. Some of the lncRNAs, their targets, and their functions are listed in Table 3.

LncRNAs are more than 200 nucleotides long and account for the majority of ncRNA [179,180]. However, less than 5% have been characterized to date, owing in part to poor conservation among species [181,182,183]. Although lncRNAs lack functional initiation codon and termination codons [184], some lncRNAs have been shown to translate into micropeptides [185]. In a study conducted by Ann et al. [186], it was shown that among the 380 RNAs that differed in expression between plaque and control tissues, lncRNA HSPA7 was increased by oxidized low-density lipoprotein (oxLDL). HSPA7 knockdown decreased human aortic smooth muscle cell migration as well as IL-1 and IL-6 secretion and expression. However, HSPA7 knockdown reversed the oxLDL-induced reduction in contractile marker expression. HSPA7 had an effect on miR-223 via an AGO2-dependent mechanism. HSPA7 is variably expressed in human atheroma and promotes transdifferentiation of contractile VSMCs phenotype to inflammatory de-differentiated/secretory phenotype through sponging miR-223. Li et al. [162], examining the serum samples of 38 patients with atherosclerosis, found that the level of the lncRNA TUG1 had dramatically increased in atherosclerotic plaques and VSMC damage models, and the expression of the lncRNA TUG1 was likewise elevated. A study by Hu et al. [187] demonstrated significant downregulation of the NEXN gene, lncRNA gene, and NEXN-AS1 in atherosclerotic lesions. An in-vivo experiment showed that the lncRNA NEXN-AS1 could increase the expression of NEXN in ECs and that NEXN-AS1 overexpression decreased endothelial inflammatory activation by blocking the NF-κB pathway [187]. It is widely accepted that oxLDL is one of the most potent inflammatory triggers for atherosclerosis and that autophagy is the survival mechanism for cells under stress. Studies demonstrated that oxLDL lowered the number and activity of mature-Cathepsin D, resulting in decreased lysosome activity, which largely contributed to impaired autophagic flux and decreased cell survival during atherogenesis [188,189].

In another study conducted by Vacante et al. [190], it was demonstrated that lncRNA CARMN and related microRNAs were downregulated in advanced versus early atherosclerotic lesions in humans and animals. Under homeostatic settings, CARMN deletion affected the expression of miR-143 and miR-145. When atherosclerosis was produced in mice, CARMN deletion increased the volume, size, and content of proinflammatory Lgals3 (galectin 3)-expressing cells and altered plaque composition, resulting in an advanced phenotype. Wang et al. [191] reported that in atherosclerotic mice and ox-LDL-stimulated VSMCs, SNHG16 and HMGB2 expression were enhanced, but the miR-22-3p expression was decreased. SNHG16 inhibited miR-22-3p expression through direct binding, and miR-22-3p mimicked reduced proliferation, migration, and invasion in ox-LDL-treated VSMCs. In addition, because HMGB2 was a target of miR-22-3p, SNHG16 increased HMGB2 levels by functioning as a competitive endogenous RNA (ceRNA) of miR-22-3p. The sh-HMGB2 inhibited ox-LDL-induced VSMC proliferation, migration, and invasion when combined with a miR-22-2p inhibitor. Through miR-22-3p/HMGB2 axis, SNHG16 accelerated atherosclerotic plaque production and increased ox-LDL-activated VSMC proliferation and migration. It has been observed that EC pyroptosis and atherosclerotic plaque formation were greatly reduced when Gaplinc was silenced. Gaplinc may interact with SP1 to bind to the NLRP3 promoter and upregulate NLRP3 target gene expression in high-fat diet-fed animals, promoting EC pyroptosis and atherosclerotic plaque growth [192]. Ni et al. [193] studied lncRNA from smooth muscle cells, which regulates cell plasticity and atherosclerosis by interacting with serum response factors. It was observed that CARMN, a lncRNA, is a key regulator of VSMC plasticity and atherosclerosis. Moreover, it was documented that in HUVECs, plasmacytoma variant translocation (PVT)1 knockdown reduced ox-LDL-induced inflammation, apoptosis, and oxidative stress. PVT1 worked as a sponge for miR-153-3p, while growth factor receptor binding protein 2 (GRB2) was identified as a miR-153-3p target. Overexpression of MiR-153-3p reduced the effects of PVT1 on ox-LDL-induced cell injury. Overexpression of GRB2 reduced the protective effects of miR-153-3p against ox-LDL-induced damage. Inhibition of PVT1 attenuated the activation of the ERK1/2 and p38 pathways via the miR-153-3p/GRB2 axis. Furthermore, in atherosclerotic mice, silencing of the PVT1 gene reduced atherosclerotic plaques, lipid formation, inflammation, oxidative stress, and apoptosis [194].

Though most atherosclerotic plaques are therapeutically silent, inflammation and persistent monocyte mobilization lead to plaque growth and instability, which might result in potentially fatal complications like myocardial infarction (MI), dementia, and brain/cerebral edema. LncRNA CCL2 controls the expression of the CCL2 gene, which codes for monocyte chemoattractant protein 1 and increases the course of vascular inflammation [195]. lncRNA NEAT1, which also interacts with a chromatin modification and decreases the production of smooth muscle cell proteins, hence promoting the phenotypic switch of VSMCs from a contractile to a synthetic state, has also been demonstrated to enhance plaque destabilization [196].

## 4. Regulation of ECM Components by ncRNAs

In the past few years, various studies have investigated the upregulation or downregulation of ncRNAs and their modulation in different animal models, clinical samples, and cell systems that mimic different diseases or diseased states to understand their specific role. The ncRNAs are directly or indirectly involved in the regulation of expression of different ECM components within the atherosclerotic plaque, discussed below. The ncRNAs regulate the gene expression of ECM proteins and different cellular processes and their modulatory effects on plaque pathogenesis through post-transcriptional and post-translational regulation (Figure 2).

The production of type III collagen is regulated by miR-29, whose target gene is COL3A. As shown in atherosclerotic mice, chronic administration of miR-29 antagonist (LNA-miR-29) results in beneficial plaque remodeling [197]. In human leiomyomas, the miR-29 effect on collagen type III expression has also been confirmed [198]. In addition, collagen type VIII may play a crucial role in the plaque destabilization process. These short collagen fibers stimulate the formation of atherosclerotic plaques by encouraging the migration and proliferation of smooth muscle cells (SMCs). In addition, apolipoprotein E (ApoE) is an endogenous inhibitor of collagen type VIII, which may explain why ApoE^−/−^ mice develop atherosclerosis. Lopes et al. [199] reported that double-knockout Col8^−/−^ ApoE^−/−^ mice display a more susceptible plaque with a thin fibrous cap than single knockout ApoE^−/−^ mice. However, type I collagen reduces arterial flexibility. The amount of miR-145 is decreased in ApoE^−/−^ mice, resulting in enhanced expression of the lysyl oxidase gene (LOX). Lysyl oxidase crosslinks collagen helices and strengthens collagen fibers, hence increasing the arterial rigidity of these mice [200]. The expression of elastin is controlled by microRNA from the miR-15 family (particularly miR-195) and the miR-29 family, and these inhibit the expression of collagen and proteoglycan. Antagomir-29b significantly reduces aortic aneurysm diameter in ApoE^−/−^ mice, whereas the miR-195 serum level corresponds with the aortic aneurysm diameter in humans [201]. Surprisingly, this is an inverse association, as miR-195 inhibits elastin and collagens and the ECM-degrading enzyme MMP-9 [201]. The molecule miR-181b is an additional epigenetic regulator of elastin gene expression [202]. In ApoE^−/−^ mice, its suppression by anti-miR-181b reduced the formation of aortic aneurysms, increased the fibrotic response, and stabilized atherosclerotic plaques or aneurysms. Decorins, a proteoglycan, are frequently used in relation to microRNA involvement. The expression of this gene is negatively regulated by miR-181b, as proven by studies on hypertrophic scars [203]. Decorin also stimulates the activation of proinflammatory macrophages via PDCD4 (programmed cell death 4) and adversely regulates miR-21 expression. Given that miR-21 is considered an oncogene (oncomir), decorin appears to inhibit cancer development [204]. However, it is thought that hyaluronic acid increases miR-10 expression. miR-10 stimulates blood vessel development by direct control of fms-related tyrosine kinase-1 (flt-1) and Mib-1 [205,206]. The significance of hyaluronic acid in the instability of atherosclerotic plaques and its regulation by microRNA molecules must be explored. Specifically, the hyaluronic acid receptor CD44 is blocked by miR-328, which has been observed in renal tubular cells [207]. Notably, proteoglycan expression can also be regulated by miR-599 in conjunction with collagen expression [208].

Peptidylarginine deiminase (PAD) plays an important role in ECM stability and remodeling. Increased levels of PAD in cardiovascular diseases (CVDs), including atherosclerosis, coronary heart disease, venous thrombosis, cardiac fibrosis, heart failure, and acute inflammation, suggesting its critical role in CVDs. PAD-mediated deamination or citrullination is involved in various physiological and pathological conditions in the body [209]. Citrullination, a post-translational process, causes the deamination of arginine (Arg) and conversion of peptidyl-based arginine to peptidyl-based citrulline. This alters the original three-dimensional structure and function of target proteins and results in dysregulated inflammatory signaling [210]. MMPs play a critical role in ECM remodeling, and along with glycosylation, nitrosylation, and proteolysis, citrullination is also involved. Hypercitrullination of MMP-9 results in a higher affinity for MMP-9 gelatin compared to control MMP-9 [211]. Further, the association of PAD-mediated citrullination of fibronectin, an important constituent of ECM, with CVDs, fibrosis, carcinogenesis, rheumatoid arthritis, alteration of integrin clustering, and focal adhesion stability suggests its role in regulating vascular remodeling because fibronectin-mediated inflammatory signaling through integrin α5 is important for vascular remodeling [212,213,214].

Collagen and elastin are the main ECM components contributing to the structural matrix and elasticity of the arteries. Collagen type I, III, IV, V, VI, XVI, XVII, nidogen, perlecan, agrin, fibronectin, laminin, and prostaglandins (PGs) are major components of the vascular wall, and type I and III fibrillar collagens, chondroitin sulfate, and dermatan sulfate PGs, and fibronectin are major ECM component in the adventitia. During remodeling, the levels of these components get altered to provide a favorable microenvironment to get a vessel to remodel during CVDs [210,215]. Various mediators regulate ECM and vascular remodeling, and post-transcriptional regulation is an important evolving aspect (Table 4). The studies presented in Table 4 suggest that IncRNAs play a regulatory role in the expression of various ECM components and the proteases modulating their expression. These findings are further supported by the involvement of ox-LDL with the inflammatory response of macrophages in atherogenesis [216], LASER, LeXis, and CHROME IncRNA in cholesterol homeostasis and foam cell formation, and MANTIS, lncRNA-CCL2, and MALAT1 in vascular inflammation [154]. Further, the functional relevance of IncRNAs with atherosclerosis [217] and the association of MALAT1, GAS5, lncRNASNP, HAND2-AS1, H19, and others, and miRNAs in atherosclerotic plaque formation [218,219] support the notion that IncRNA plays an important role in atherosclerotic plaque formation and progression. Moreover, the regulation of smooth muscle cell proliferation and calcification plays a critical role in plaque formation and regulation of MMP-16, co-expressed with MMP-2 and MMP-9 and various other MMPs by IncRNAs [220]. All these effects support the role of and warrant a further in-depth understanding of the role of lncRNA-mediated regulation of plaque formation and progression, ECM and vascular remodeling, and associated complications.

## 5. Translational Aspects and Clinical Significance

As discussed above, the expression levels of various components of ECM are regulated by ncRNA. However, the research studies investigating this correlation are limited in the literature. The available studies and clinical trials (NCT03603431, NCT03494712, NCT02603224, and NCT04045405) [237] have discussed the role of miR-92a, miR-29b, and miR-132 in association with cutaneous healing and cardiac fibrosis, both having similar pathogenesis of inflammation and ECM remodeling. This implies that these miRNAs may also regulate ECM remodeling during plaque formation and progression, an inflammatory pathology of the vessels. This notion is supported by the fact that miR-92a is involved in angiogenesis, vascular inflammation, and vasodilation; miR-29b regulates elastin degradation; miR-132 regulates vascular smooth muscle cell proliferation and neointimal hyperplasia [66,67,99,102,103]. Although the studies investigating ncRNA-mediated ECM remodeling are limited, the involvement of ncRNAs regulating molecular mechanisms in plaque pathogenesis warrants further research. In the context of the treatment of plaque pathology, preclinical investigations have proven that several methods have a plaque-stabilizing impact by targeting apolipoprotein E, apolipoprotein B, and LDLs in SMCs, macrophages, monocytes [238,239,240,241]. Most of these studies are in animal models; thus, the positive outcomes have not been replicated in human clinical trials [242,243,244,245,246]. This may be due to different molecular compositions (macrophage subsets), locations, pathophysiological processes involved in atherosclerotic plaque instability, the animal model used, and varying human populations [247,248,249].

The clinical trials conducted in the treatment of atherosclerosis are mainly focused on the outcomes of cardiovascular diseases and acute ischemic events. Canakinumab administered to individuals with a prior myocardial infarction resulted in a substantial decline in subsequent cardiovascular problems in comparison to placebo [250]. Accordingly, in the COLCOT study [251], colchicine administered to patients after a myocardial infarction resulted in a considerable decline in composite endpoint and a significant reduction in recurrent myocardial infarction. In comparison, an experiment called STABILITY with darapladib, which was performed on patients with stabilized cardiac artery disorder (no prior myocardial infarction), was unable to show a statistically considerable difference between the darapladib and placebo groups in terms of composite endpoint and mortality, despite showing a subtle but notable decline in significant cardiac problems [252]. Similarly, the cholesterylester transfer protein (CETP) inhibitor anacetrapib, which causes an increase in HDL, showed a minor but substantial reduction in major coronary events [253]. The results from these and other clinical trials (Table 5) suggest that these drugs mainly stabilize plaque or attenuate atherosclerosis and target the ncRNA involved in ECM remodeling, inflammation, stabilization of atherosclerotic plaque, or other related events will be of significance in the treatment of atherosclerosis. Of note, to determine whether a specific type of therapy results in atherosclerotic plaque stability, the composition and morphology of the plaque must be visualized, and their stability exponents must be assessed using intravascular ultrasonography and optical coherence tomography [254,255] to enhance the therapeutic efficacy of the agent under consideration.

A convergence of basic and clinical research has significantly transformed the strategies for managing atherosclerosis and involves mainly targeting inflammatory components. This was mainly due to the advancement in the approach of randomized clinical trials involving individuals with an atherosclerotic plaque at different stages and treatment strategies. Furthermore, understanding plaque pathology has also been aided by improvements in human genetic studies enabled by next-generation sequencing and other technological innovations, along with an ever-evolving toolbox in the form of genetically modified mice models allowing for gene-editing and induced pluripotential stem cell methodology [267]. Understanding the activities of ncRNAs in atherosclerosis has progressed beyond DNA and mRNA analyses because of the involvement of microRNAs and lncRNAs regulating gene transcription in atherosclerosis [67,268].

## 6. Conclusions

Based on the studies discussed in this article, it is evident that ECM remodeling is epigenetically regulated involving miRNAs, lncRNA, and circRNA, and these ncRNAs regulate the expression of various proteins involved during plaque formation and vulnerability. Since ECM remodeling plays a critical role in plaque vulnerability to stabilize plaque, ncRNAs can be strong contenders to target. Additionally, the levels of these change during the process of plaque formation, as evidenced by various studies. ncRNAs may also serve as diagnostic and prognostic biomarkers for atherosclerosis. Therefore, ncRNAs can be strong contenders for therapeutic targets for atherosclerosis and related disorders, and the identification and characterization of relative ncRNAs may have clinical applications, both as prognostic tools and for therapeutic targets. Further investigations are required to develop and use specific ncRNAs in diagnosis and therapeutics in patients with cardiovascular diseases. Translating these scientific advancements in therapeutics has necessitated large-scale clinical trials, which have necessitated increased creativity and money due to the success of conventional treatments. Placebo-controlled, randomized clinical trials continue to be the most reliable approach for evaluating the applicability of lab findings to patients. Indeed, the globalization of cardiovascular disease risk has raised the overall burden of atherosclerotic disease. However, the progress in laboratory and clinical research promises to provide us with methods to combat this global epidemic. To make progress in the control of atherosclerosis, a multidisciplinary collaboration of public health measures, applied behavioral psychology, risk factor control, consistent implementation of existing therapies, and the development and validation of new therapeutic approaches will be required.

## Figures and Tables

**Figure 1 ijms-23-13731-f001:**
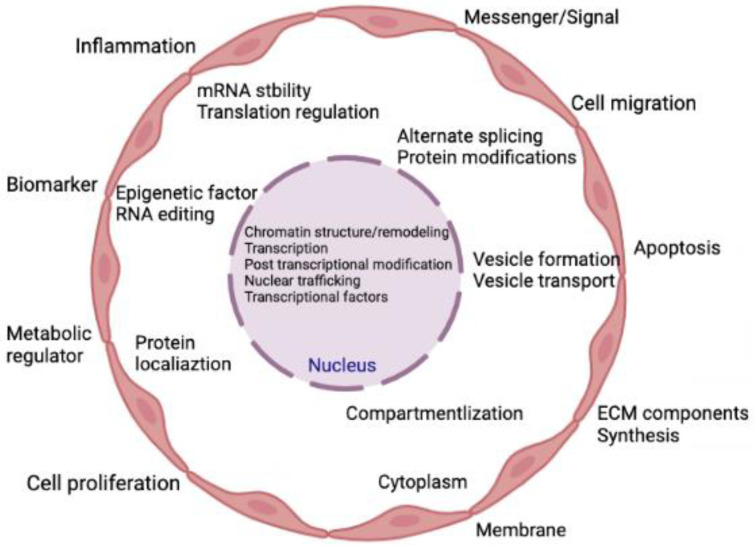
Summary of the cellular activities that are regulated by ncRNAs. Indeed, ncRNAs can directly and simultaneously modulate multiple targets and are involved in both gene expression and genome remodeling. Thus, ncRNAs control cellular functions directly or indirectly in both physiological and pathological conditions.

**Figure 2 ijms-23-13731-f002:**
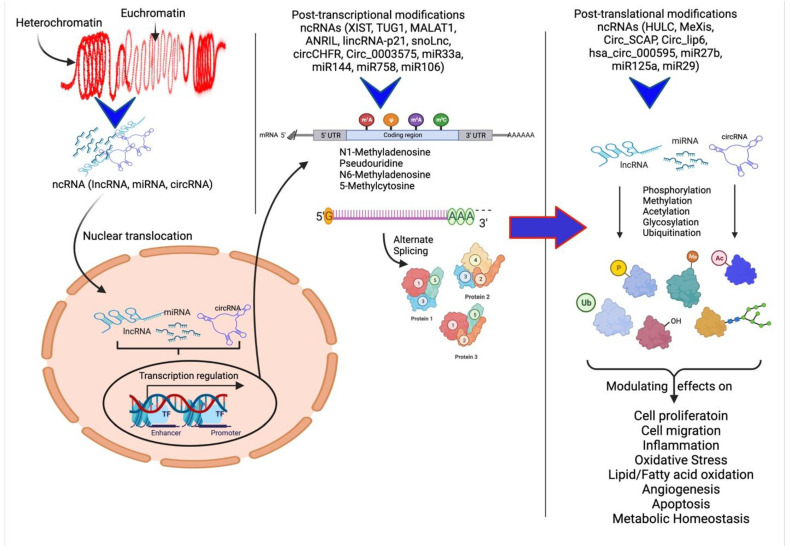
**Schematic representation of ncRNAs acting on various cellular processes and their modulatory effects.** Non-coding RNAs (ncRNAs) regulate gene expression at the transcriptional and post-transcriptional levels and are also involved in the epigenetic regulation of various genes. The ncRNAs play a critical role in heterochromatin formation and histone modification involving methylation, acetylation, ubiquitination, citrullination, alternative splicing, and gene silencing. Modified protein structure and levels regulate various molecular mechanisms involved in angiogenesis, cell proliferation and migration, inflammation, and remodeling.

**Table 1 ijms-23-13731-t001:** miRNAs, their target, and functions.

Type of ncRNA	Target RNA	Function	References
miR-34a	Sirtuin 1 (SIRT 1)	Contractile function, apoptosis	[65]
miR-92a	Kruppel-like factor 4 (KLF4)	Inhibits angiogenesis	[66,67]
miR-126	Intercellular adhesion molecule 1 (ICAM-1), Vascular cell adhesion molecule 1 (VCAM-1)	Regulation of inflammation promotes plaque regression	[68,69]
miR-27b	Peroxisome Proliferator-Activated Receptor (PPAR) Gamma (PPARγ), Angiopoietin-like 3 (Angptl3), mitochondrial Glycerol-3-Phosphate Acyltransferase (Gpam)	Plaque progression and development	[70,71]
miR-143/145	PPARγ, Angptl3, Gpam	Maintain VSMC contractile phenotype	[72]
miR-21	Phosphatase and Tensin Homolog (PTEN)/v-Akt Murine Thymoma Viral Oncogene (AKT)/Extracellular signal-regulated kinase (ERK) regulation	Promotes contractile phenotype	[73]
miR-125a-5p	Oxysterol-binding protein (OSBP)-related protein 9 (ORP-9)	Inhibits proinflammatory signals	[74,75]
miR-146	Toll-like receptor 4 (TLR4)	Inhibits proinflammatory signals	[76]
miR-33a/b	ATP Binding Cassette (ABC) Subfamily A Member 1 (ABCA1), ABC Subfamily G Member 1 (ABCG1), Carnitine Palmitoyl transferase 1A (CPT1A), Carnitine O-Octanoyl transferase (CROT), Hydroxy acyl-CoA Dehydrogenase Trifunctional Multienzyme Complex Subunit Beta (HADHB)	Cholesterol efflux, fatty acid β-oxidation	[77,78]
miR-144, miR-758, miR-106	ABCA1	Cholesterol efflux	[79,80]
miR-30c	Microsomal triglyceride transfer protein (MTP), Lysophosphatidyl glycerol Acyltransferase 1 (LPGAT1)	Cholesterol synthesis, lipoprotein secretion	[81]
miR-155	LX1, Cluster of differentiation (CD) 36 (CD36), CD68, Myeloid differentiation primary response 88 (MyD88), B-Cell lymphoma 6 (BCL6)	Lipid uptake and inflammation	[82,83]
miR-125a-5p	ORP9	Lipid uptake and inflammation	[74]
miR-146a	TLR4	TH1 response	[76]
miR-9	PPARδ	Inflammation	[84]
miR-21	Tropomyosin 1(TPM1), Programmed Cell Death 4 (PDCD4), PPARα	Proliferation, migration, and apoptosis	[85]
miR-143/145	KLF4, KLF5, ETS domain transcription factor 1 (ELK-1)	Phenotype switching, podosome formation	[86,87]
miR-21	TPM1, PDCD4, PPARα	Proliferation, migration, and apoptosis	[85,88]
miR-1/33	KLF4, Specificity protein 1 (Sp-1)	Proliferation	[89,90]
miR-221/222	Cyclin-dependent kinase inhibitor (CDKN) 1B (p27), CDKN1C (p57), Tyrosine protein kinase c-KIT (CD117)	Proliferation, migration, and apoptosis	[91,92]
miR-29	Elastin	Elastin formation	[93,94]
miR-208	CDKN1A (p21)	Proliferation	[95]
let-7d	Kirsten rat sarcoma virus (KRAS)	Proliferation	[96]
let-7 g	Lectin-type oxidized LDL receptor 1 (LOX-1)	Proliferation and migration	[97,98]
miR-132	Leucine-rich repeat flightless-interacting protein 1 (LRRFIP1)	Proliferation	[99]
miR-133a	Runt-related transcription factor 2 (RUNX2)	Osteogenic differentiation	[100]
miR-126	Sprouty-related EVH1 domain-containing protein 1 (SPRED1), VCAM-1	Monocyte adhesion	[68]
miR-17-3p, miR-31	ICAM-1, E-selectin	Inflammation	[101]
miR-92a	Endothelial nitric oxide synthase (eNOS), KLF2, KLF4, Suppressor of cytokine signaling 5 (SOCS5)	vasodilation, inflammation	[102,103]
miR-155, miR-221/222	eNOS, ETS Proto-Oncogene 1 (ETS1)	Inflammation	[104,105]
miR-712	Tissue inhibitor of metalloproteinase 3 (TIMP3)	Inflammation	[106]
miR-10	VCAM-1, E-selectin	Inflammation	[107]
miR-181b	Importin subunit alpha 3 (Importin α3)	Inflammation	[68]
miR-27	Semaphorin 6A (SEMA6A)	EC adhesion, angiogenesis	[108]
miR-34a, miR-217	SIRT-1	Senescence	[109]
miR-146	Human Antigen R (HuR), Reduced nicotinamide adenine dinucleotide phosphate (NADPH) Oxidase 4 (NOX4)	EC activation, aging	[110]

**Table 2 ijms-23-13731-t002:** Circular RNAs, their targets, and functions.

Type of circRNA	Target RNA	Function	Reference
CircANRIL	N/A	Apoptosis, inhibits proliferation	[122]
Has_circ_0010729	Hypoxia-inducible factor 1-alpha (HIF-1α)	Cell proliferation, and migration, inhibits apoptosis	[123]
cZNF609	Myocyte Enhancer Factor 2A (MEF2A)	Apoptosis, inflammation, Inhibits proliferation and migration	[124]
circRELL1	MyD88/Nuclear factor kappa B (NF-kB)	Increases Inflammation	[125]
Circ_CHFR	Forkhead Box O1 (FOXO1), Cyclin D1 (CCND1)	Cell proliferation and migration	[126]
Circ-SATB2	Stromal Interaction Molecule 1 (STIM1)	Cell proliferation and migration, inhibit apoptosis	[127]
CircWDR77	Fibroblast growth factor 2 (FGF2)	Cell proliferation and migration	[128]
CircTM7SF3	Aspartate Beta-Hydroxylase (ASPH)	Apoptosis, inflammation, oxidative stress	[129]
CircSCAP/has_circ_0001292	Phosphodiesterase 3B (PDE3B)	Accumulation of lipids, inflammation, and oxidative stress	[130]
has_circ_0054633	Roundabout homolog 1 (ROBO1) and Heme Oxygenase 1 (HO-1)	Cell proliferation, migration, angiopoiesis, apoptosis inhibition	[131]
CircHIPK3/mmu_circ_0001052	Frizzled Class Receptor 4 (FZD4) and Wingless family member 2 (WNT2)	Cell proliferation and inflammation	[132]
CircHIPK3	Insulin-like growth factor 1 (IGF-1)	Apoptosis and oxidative stress inhibition	[133]
CircDNMT3B	Bone morphogenetic protein (BMP) And Activin Membrane Bound Inhibitor (BAMBI)	Cell proliferation and migration	[13]
Circ_0003575	Forkhead Box O3 (FOXO3), Forkhead Box O4 (FOXO4)	Cell proliferation and migration, inhibit apoptosis	[134]
Hsa circ 0068087	TLR4	Increases Inflammation	[135]
CircRNA-0044073	Janus kinase/signal transducers and activators of transcription (JAK/STAT)	Cell proliferation and migration	[136]
hsa_circ_0003575	miR-9, miR-199	Cell proliferation, angiogenesis	[137]
hsa_circ_000595	miR-19a	Apoptosis	[138]
Circ_Lip6	miR-145	Cell proliferation and migration	[139]

**Table 3 ijms-23-13731-t003:** Long noncoding RNAs, their targets, and functions.

Type of lncRNA	Target RNA	Function	Reference
Lnc-Ang362	miR-221/222	Proliferation	[155]
HIF-AS1	Cholecystokinin-8 (CCK-8)	Apoptosis, inhibits proliferation	[156,157]
HULC	DNA (cytosine-5)-methyltransferase 1 (DNMT1)	Apoptosis	[158]
lincRNA-p21	Tumor protein 53 (p53), Mouse double minute 2 homolog (MDM2)	Apoptosis, inhibits proliferation	[159,160]
TUG1	miR-62, miR-21, Phosphatase and Tensin Homolog (PTEN)	Apoptosis, Cell proliferation	[161,162]
MALAT1	C-X-C Motif Chemokine Receptor 2 (CXCR2)	Apoptosis, inflammation, inhibits proliferation	[163]
MeXis	Abca1	Lipid metabolism, inflammation	[164]
H19	Wingless family member 1 (WNT1)	Apoptosis, inhibits proliferation	[165]
DIL4-AS	CD31, Hairy, and enhancer of split-1 (HES1)	Cell proliferation and migration	[166]
GAS5	Matrix metalloproteinases (MMPs), High Mobility Group Box 1 (HMGB1)	Inflammation, apoptosis	[167]
MIAT	STAT3	Cell proliferation, inhibits apoptosis	[168,169]
SENCR	Myocardin (Myocd), Midkine (MDK), and pleiotrophin (PTN)	Cell proliferation and migration	[170,171]
XIST	Nucleotide-binding oligomerization domain-containing protein 2 (NOD2)	Apoptosis	[172]
sONE	eNOS, *c-myc*	Inhibits cell proliferation	[173]
MEG3	NLR family pyrin domain containing 3 (NLRP3), Ras Homolog Family Member B (RhoB)/PTEN	Inflammation, proliferation	[174,175,176]
ANRIL	CDKN2A (p16)	Cell proliferation	[141,177]
SIRST1 antisense	SIRT1	Cell proliferation and migration	[178]

**Table 4 ijms-23-13731-t004:** Various ECM components regulated by ncRNAs.

Protein	ncRNA	Strategy	Outcome
Collagen [221]	lncRNA8975-1	In-vitro studies to investigate the effects of overexpression and knockdown of lncRNA8975-1 on collagen expression in dermal fibroblasts.	lncRNA8975-1 was overexpressed in hypertrophic scar tissues and dermal fibroblasts lncRNA8975-1 regulates the protein expression levels of COL1A2, COL1A1, COL3A1
Collagen [222]	LncRNA AC067945.2	In-vitro studies to investigate the effects of overexpression of LncRNA AC067945.2 on collagen expression in normal skin fibroblasts.	LncRNA AC067945.2 overexpression inhibits the expression of COL1A1, COL1A2, COL3A1 LncRNA AC067945.2 represses VEGF secretion
Collagen [223]	lncRNA TP53TG1	In vivo (6 weeks old C57BL/6 male mice) overexpression of TP53TG1 by adeno-associated virus 5 to examine its effect on idiopathic pulmonary fibrosis	Overexpression of TP53TG1 attenuates the increased expression of FN1, Col1α1, Col 3α1, ACTA2 mRNA, FN1, and Col I protein level
Collagen [224]	LncRNA SCARNA10	In vitro and in vivo (Balb/c mice) evaluation of the effects of LncRNA SCARNA10 overexpression and knockdown on fibrosis	High expression of SCARNA10 is positively associated with Col1α1 expression
Collagen [225]	shRNA-NEAT1-1	To investigate the roles of LncRNA NEAT1 and microRNA-455-3p in pulmonary fibrosis using alveolar epithelial cells	shRNA-NEAT1-1 abrogates the promotional effects of TGF-β1 on the protein expression levels of collagen I and III and regulate pulmonary fibrosis involving microRNA-455-3p/SMAD3 axis
Collagen [226]	LncRNA PVT1	Evaluated the role of LncRNA PVT1 in atrial fibrosis in Ang-II-treated human atrial fibroblasts and Ang-II-induced atrial fibrosis in mice	Increased expression of LncRNA PVT1 is positively associated with Col I and Col III Regulates atrial fibrosis via miR-128-3p-SP1-TGF-β1-Smad axis in atrial fibrillation
Collagen [227]	Lnc-HOTAIR	Role of Lnc-HOTAIR in gastric cancer growth and metastasis	Lnc-HOTAIR is positively associated with COL5A1 expression by sponging miR-1277-5p
Fibronectin and Myh-11 [228]	lncRNA-AK098656	LncRNA microarray and whole-genome microarray in human plasma samples and transgenic rats	Highly upregulated in the plasma of hypertensive patients and predominantly expressed in VSMC Binds to myosin heavy chain-11 and FN1 and promotes degradation Promote synthetic phenotype in VSMCs
Elastin [229]	lncRNA TUG1	Investigated the role and mechanism of lncRNA TUG1 in bronchopulmonary dysplasia using a mouse model	lncRNA TUG1 negatively regulates miR-29a-3p miR-29a-3p negatively regulates elastin lncRNA TUG1 suppresses the inflammatory response and cell apoptosis
Collagen and fibronectin [230]	IncRNAs (n379599, n379519, n384648, n380433, and n410105)	RNA deep sequencing of protein-coding and non-coding RNAs from cardiac samples of patients with ischemic cardiomyopathy and cardiac fibroblasts from the mouse were used	lncRNA expression is positively associated with the expression of COL3A1, COL8A1, and FN1
Collagen [231]	lncRNA GATA6-AS	Investigated the role of hypoxia-responsive lncRNA GATA6-AS in endothelial cells growth and proliferation by RNA sequencing using HUVECs	Acts as a negative regulator of nuclear LOXL2 function GATA6-AS regulates H3K4me3 methylation of periostin and cyclooxygenase-2 Collagen IV scaffolding is inversely regulated by LOXL2 and GATA6-AS silencing GATA6-AS is upregulated in endothelial cells during hypoxia
Collagens and elastin PMID: 33473324	lncRNA Cfast	Investigated the role of lncRNA Cfast in cardiac fibrosis	Inc RNA Cfast is positively associated with Col1α1, Col3 α 1, elastin, and α-SMA expression, and depletion of Cfast attenuate their expression
MMP-9 [232]	lncRNA LINC00460	Investigated the role of lncRNAs in meningioma using human tissues and meningioma cell line (Ben-Men-1)	LINC00460 is positively associated with MMP-2 and MMP-9 expression LINC00460 promotes MMP-9 expression by targeting miR-539
MMP-16 [233]	LncRNA NEAT1	To investigate the role of LncRNA NEAT1 in regulating inflammation in asthma using BEAS-2B cells	NEAT1 negatively regulates miR-200a/b expression MMP-16 is a target gene of miR-200a/b
MMP-2 [234]	lncRNA GAS5	Investigating the effect and mechanism of lncRNA GAS5 in cardiac fibrosis using C57BL/6 mice	lncRNA GAS5 was significantly downregulated in cardiac fibrosis Overexpression of GAS5 decrease MMP-2 and Col 1
MMP-1 [235]	LncRNA WTAPP1	Investigated the role of LncRNA WTAPP1 in the regulation of efficient recruitment and angiogenesis of endothelial progenitor cells (EPCs; in-vitro)	WTAPP1 positively regulated migration, invasion, and tube formation in EPCs by increasing MMP-1 expression and activating PI3K/Akt/mTOR signaling. lncRNA WTAPP1 is a molecular decoy for miR-3120-5p
MMP-9 [236]	TET2-interacting long noncoding RNA (TETILA)	To investigate the interaction between demethylation enzymes like TET2 with lncRNA to target specific promoters	TETILA indirectly activates MMP-9 promoter demethylation

**Table 5 ijms-23-13731-t005:** Clinical trials in atherosclerosis.

Trial Name, Acronym	Intervention	Outcome	References
Anti-inflammatory therapy for CAD	Placebo vs. colchicine	Reduces the risk of recurrent myocardial infarction	[256]
Aggressive Reduction of inflammation stops events	Succinobucol vs. placebo	Succinbucol did not affect the primary endpoint	[257]
Anti-inflammatory medications on cardiovascular outcomes of coronary artery disease patients	Pexelizumab, anakinra, colchicine, darapladib, varespladib, canakinumab, inclacumab, and losmapimod	Failed to reduce adverse cardiovascular outcomes	[258]
Investigation of Lipid Level Management to Understand Its Impact on Atherosclerotic Events	Torcetrapib + atorvastatin vs. atorvastatin	Torcetrapib increased HDL levels, decreased LDL levels, increased blood pressure, increased cardiovascular mortality	[259]
Association between bleeding and subsequent major adverse cardiac and cerebrovascular events (MACCE)	Rivaroxaban or rivaroxaban plus an antiplatelet agent	In patients with atrial fibrillation and stable coronary artery disease, major bleeding was strongly associated with subsequent MACCE	[260]
A Study of RO4607381 in Stable Coronary Heart Disease Patients With Recent Acute Coronary Syndrome	Optimal medical therapy—dalcetrapib vs. optimal medical therapy + placebo	Dalcetrapib increased HDL levels but did not reduce cardiovascular events	[261]
anti-inflammatory agents in CAD	Colchicine	Gout patients who took colchicine had a significantly lower prevalence of myocardial infarction and reduced all-cause mortality and CRP level	[262]
Randomized Evaluation of the Effects of Anacetrapib Through Lipid-modification	Anacetrapib vs. placebo	Not published yet	[263]
anti-inflammatory agents in CAD	Colchicine	Reduced risk of a CV event among patients with gout.	[264]
Stabilization Of Atherosclerotic Plaque By Initiation of DarapLadIb Therapy	Optimal medical therapy + Darapladib vs. optimal medical therapy + placebo	Darpladib did not reduce the composite endpoint	[265]
Efficacy of Pioglitazone on Pacrovascular Outcome in Patients with Type 2 Diabetes	Pioglitazone vs. Placebo	Pioglitazone reduces the composite of all-cause mortality, non-fatal myocardial infarction, and stroke in patients with type 2 diabetes	[266]
Rosiglitazone evaluated for cardiovascular outcomes in oral agent combination therapy for type 2 diabetes (RECORD): a multicenter, randomized, open-label trial	Rosiglitazone vs. Placebo	Rosiglitazone does not increase the risk of overall cardiovascular morbidity or mortality compared with standard glucose-lowering drugs	

CAD, coronary artery disease; CV, cardiovascular; CRP, C-reactive protein; HDL, high-density lipoprotein; LDL, low-density lipoprotein.

## Data Availability

Not applicable since the information is gathered from published articles.

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
