# Peer review of "Non-Coding RNAs in Regulating Plaque Progression and Remodeling of Extracellular Matrix in Atherosclerosis"

_ijms, 2022, doi:10.3390/ijms232213731_

Round 1
Reviewer 1 Report
In this review entitled “Non-coding RNAs in regulating plaque progression and remodeling of extracellular matrix in atherosclerosis”, Singh et al. provide a comprehensive review on the ncRNA-associated mechanisms of atherosclerosis and related cardiovascular disorders with particular emphasis on the regulation of ECM remodeling by various ncRNA molecules, because de-regulated ECM remodeling contributes to the development of atherosclerosis.
The discovery of non-coding ribonucleic acids (ncRNAs) altered the understanding of the post-translational, post-transcriptional, and epigenetic regulation of gene expression in controlling cellular homeostasis in various diseases. However, currently little is known about the ncRNA-associated mechanisms of atherosclerosis and related cardiovascular disorders. An in-depth understanding of ncRNA-associated ECM remodeling may help to identify novel targets for the treatment of atherosclerosis and other cardiovascular diseases. In summary, this article provided a timely and interesting review on ncRNA associated mechanisms of atherosclerosis and related cardiovascular disorders.
The following are minor concerns, which need to be modified to improve clarity and presentation.
1. Some citations might be inaccurate. The authors need to check the accuracy of their citations. For example, citations 1-4 need to be corrected.
2. An illustration of Non-Coding RNAs will enhance this review.
Author Response
Response to the Comments of Reviewer #1:
Comments: In this review entitled “Non-coding RNAs in regulating plaque progression and remodeling of extracellular matrix in atherosclerosis”, Singh et al. provide a comprehensive review on the ncRNA-associated mechanisms of atherosclerosis and related cardiovascular disorders with particular emphasis on the regulation of ECM remodeling by various ncRNA molecules, because de-regulated ECM remodeling contributes to the development of atherosclerosis. The discovery of non-coding ribonucleic acids (ncRNAs) altered the understanding of the post-translational, post-transcriptional, and epigenetic regulation of gene expression in controlling cellular homeostasis in various diseases. However, currently little is known about the ncRNA-associated mechanisms of atherosclerosis and related cardiovascular disorders. An in-depth understanding of ncRNA-associated ECM remodeling may help to identify novel targets for the treatment of atherosclerosis and other cardiovascular diseases. In summary, this article provided a timely and interesting review on ncRNA associated mechanisms of atherosclerosis and related cardiovascular disorders. The following are minor concerns, which need to be modified to improve clarity and presentation.
Response: Thank you for your encouraging comments!
Concern 1: Some citations might be inaccurate. The authors need to check the accuracy of their citations. For example, citations 1-4 need to be corrected.
Response: We have corrected the references.
Concern 2: An illustration of Non-Coding RNAs will enhance this review.
Response: We have added the information and accordingly reorganized the review as per the suggestion.
Reviewer 2 Report
The manuscript describes the contribution of ncRNA (miRNA, cRNA and lncRNA) in the atherosclerotic process.
The article has abundant information. However, there are several articles of non-coding RNA associated with atherosclerosis and the section of “Regulation of ECM components by ncRNA” did not present any novelty. So, what is the relevance of this article?
Author Response
Response to the Comments of Reviewer #2:
Comments: The manuscript describes the contribution of ncRNA (miRNA, cRNA and lncRNA) in the atherosclerotic process. The article has abundant information.
Response: Thank you for your encouraging comments!
Concern: However, there are several articles of non-coding RNA associated with atherosclerosis and the section of “Regulation of ECM components by ncRNA” did not present any novelty. So, what is the relevance of this article?
Response: We have modified the section of “Regulation of ECM components by ncRNAs”. We have included the most recent information on this topic. Also, we have specifically discussed the individual ECM components and their regulation which was not discussed in any of the previous articles.
Reviewer 3 Report
The manuscript of Singh et al intends to review an interesting aspect of atherosclerosis, remodelling of extracellular matrix (ECM). However, most of the manuscript is centred towards ncRNAs and very little regards ECM remodelling and the role of ncRNAs in its regulation. It is not clear the design of this review. In this form it is not acceptable for publication.
This manuscript requires some important revisions:
1. The general outline of the review must be changed.
2. A brief summary of the role of remodelling of extracellular matrix in atherosclerosis must be added in the Introduction section.
3. I think that paragraph “Extracellular matrix” (paragraph 3) must be first of “non coding RNA” (Paragraph 2).
3. The paragraph “Regulation of ECM components by ncRNAs” must be expanded! This is the most interesting part of the proposed review! In this paragraph, the description of some important constituent of ECM remodelling such as PAD (peptidylarginine deiminases) enzymes are missing. In addition, it is insufficient in describing ncRNAs regulating the principal ECM proteins; i.e. collagen, that is the most abundant matrix protein. Collagen consists of 28 family members identified to date. 13 of these are found in the vascular wall, synthesised by endothelial cells, VSMCs and fibroblasts. In this paragraph are cited only collagen type III and type VIII…. The change in the composition of the ECM by matrix metalloproteinases (MMPs) is critical; also, these factors, and their regulatory ncRNAS, must be deeply discussed. Moreover, indication of ncRNAs targeting some of the principal ECM proteins such as fibronectins and integrins are lacking. Why did the authors not describe each group of ncRNAs (miRNAs, cricRNA, lnRNA) in ECM remodelling as they did for ncRNA in atherosclerosis (from page 3 to page 10!)?
4. Add a Figure and a Table representing ncRNAs acting on ECM components
5. The paragraph “Translational aspects and clinical significance” is confusing. I suggest describing first the clinical trials on ncRNAS acting on ECM components and thus regulating the remodelling of ECM. After this description, a detailed comparison with drugs used for the treatment of atherosclerosis must be added.
MINOR REVISION
1. I do not understand the meaning of sentence at page 2 lines 51-53. Please write in a clear way.
2. Modify sentence pag 2 lines 62-63. To write “only 1% of genes are coding….” is confusing.
3. Page 2 line 73 add other review as reference for the involvement of ncRNAs in pathological processes; for example: Bhattacharyya N et al., 2021; Rizzacasa et., 2019; Gorabi AM et al., 2021; Colpaert RMW et al., 2019 ecc…..
4. Page 3 line 93: check % of genes regulated by miRNAs and add a more appropriate reference.
5. Pag.10 lines 292-293: re-write the sentence. It seems that ECM constitutes only atherosclerotic plaque composition.
6. pag. 10 lines 294-309: these sentences are too similar to introduction of paper of Gialeli et al (Curr Opin Lipidol, 2021 Oct 1;32(5):277-285). Re-write!
7. pag. 11 line 337: check thickness of TFCA’s fibrous cap…mm not mm!!!!!!!
8. Each acronym must be described the first time it is cited in the text. For example, CETP at page 12 line 410.

Author Response
Response to the Comments of Reviewer #3:
Comments: The manuscript of Singh et al intends to review an interesting aspect of atherosclerosis, remodelling of extracellular matrix (ECM). However, most of the manuscript is centred towards ncRNAs and very little regards ECM remodelling and the role of ncRNAs in its regulation. It is not clear the design of this review. In this form it is not acceptable for publication.
Concern 1. The general outline of the review must be changed.
Response: The general outline of the review has been modified according to the suggestion.
Concern 2. A brief summary of the role of remodelling of extracellular matrix in atherosclerosis must be added in the Introduction section.
Response: We have explained the role of remodelling of ECM in atherosclerosis in the Introduction section.
Concern 3. I think that paragraph “Extracellular matrix” (paragraph 3) must be first of “non coding RNA” (Paragraph 2).
Response: We have rearranged the paragraphs.
Concern 4. The paragraph “Regulation of ECM components by ncRNAs” must be expanded! This is the most interesting part of the proposed review! In this paragraph, the description of some important constituent of ECM remodelling such as PAD (peptidylarginine deiminases) enzymes are missing. In addition, it is insufficient in describing ncRNAs regulating the principal ECM proteins; i.e. collagen, that is the most abundant matrix protein. Collagen consists of 28 family members identified to date. 13 of these are found in the vascular wall, synthesised by endothelial cells, VSMCs and fibroblasts. In this paragraph are cited only collagen type III and type VIII…. The change in the composition of the ECM by matrix metalloproteinases (MMPs) is critical; also, these factors, and their regulatory ncRNAS, must be deeply discussed. Moreover, indication of ncRNAs targeting some of the principal ECM proteins such as fibronectins and integrins are lacking. Why did the authors not describe each group of ncRNAs (miRNAs, cricRNA, lnRNA) in ECM remodelling as they did for ncRNA in atherosclerosis (from page 3 to page 10!)?
Response: We have made the changes as per the given suggestions.
Concern 5. Add a Figure and a Table representing ncRNAs acting on ECM components
Response: We have added a table and a figure in the revised manuscript.
Concern 6. The paragraph “Translational aspects and clinical significance” is confusing. I suggest describing first the clinical trials on ncRNAS acting on ECM components and thus regulating the remodelling of ECM. After this description, a detailed comparison with drugs used for the treatment of atherosclerosis must be added.
Response: We have updated the manuscript according to the suggestions.
MINOR REVISION
Concern 1. I do not understand the meaning of sentence at page 2 lines 51-53. Please write in a clear way.
Response: The sentence has been modified.
Concern 2. Modify sentence pag 2 lines 62-63. To write “only 1% of genes are coding….” is confusing.
Response: We have changed the % of genes to Approx. 1-2% of genes.
Concern 3. Page 2 line 73 add other review as reference for the involvement of ncRNAs in pathological processes; for example: Bhattacharyya N et al., 2021; Rizzacasa et., 2019; Gorabi AM et al., 2021; Colpaert RMW et al., 2019 ecc…..
Response: The references suggested by the reviewer have been added in the revised manuscript. (PMID: 31323768, PMID: 31488273, PMID: 34849161, PMID: 33494668)
Concern 4. Page 3 line 93: check % of genes regulated by miRNAs and add a more appropriate reference.
Response: There are more than 60% of human protein coding genes that have miRNA target sites in their 3’-UTR and various studies have shown the involvement of miRNA/mRNA interactions as key regulatory network in different biological processes. (PMID: 18955434, PMID: 30108335, PMID: 33452500, PMID: 30469501, PMID: 22436747, PMID: 22436491)
Concern 5. Pag.10 lines 292-293: re-write the sentence. It seems that ECM constitutes only atherosclerotic plaque composition.
Response: The sentence has been rewritten.
Concern 6. pag. 10 lines 294-309: these sentences are too similar to introduction of paper of Gialeli et al (Curr Opin Lipidol, 2021 Oct 1;32(5):277-285). Re-write!
Response: This whole paragraph has been rewritten.
Concern 7. pag. 11 line 337: check thickness of TFCA’s fibrous cap…mm not mm!!!!!!!
Response: mm has been changed to µm.
Concern 8. Each acronym must be described the first time it is cited in the text. For example, CETP at page 12 line 410.
Response: CETP has been described in the text.
Reviewer 4 Report
In this review paper, the authors aim to offer an up-to-date review of noncoding RNAs (miRNAs, lncRNA, and circRNA) in cardiovascular diseases. According to the studies discussed in this article, it is evident that the extracellular matrix is regulated by several miRNAs, lncRNA, and circRNA. Besides, these ncRNAs can be strongly considered diagnostic and prognostic biomarkers. There are lots of translational research value of these ncRNAs in the therapeutics of Atherosclerosis. This is a very valuable review paper for atherosclerosis research.
Author Response
Response to the Comments of Reviewer #4:
Comments: In this review paper, the authors aim to offer an up-to-date review of noncoding RNAs (miRNAs, lncRNA, and circRNA) in cardiovascular diseases. According to the studies discussed in this article, it is evident that the extracellular matrix is regulated by several miRNAs, lncRNA, and circRNA. Besides, these ncRNAs can be strongly considered diagnostic and prognostic biomarkers. There are lots of translational research value of these ncRNAs in the therapeutics of Atherosclerosis. This is a very valuable review paper for atherosclerosis research.
Response: Thank you for your encouraging comments.
We hope that the revised manuscript is now suitable for publication in IJMS.
Round 2
Reviewer 2 Report
The manuscript has some mistakes.
Author Response
Response to the Comments of Reviewer #2:
Comments: The manuscript has some mistakes.
Response: Thank you! The manuscript has been carefully edited to correct any error.
Reviewer 3 Report
I noticed that the authors have made all the changes I outlined and modified the first 3 references; I think these new references are more appropriate.
However, the authors must check accurately the reference list and the references cited in the text.
1. For example, the reference 13 (Zhu K et al., 2019) refers to diabetic diseases and ref 16 (Zhang F et al., 2018) to cardiovascular disease. However, in the text ref 13 is linked to CVD and 16 to diabetes!
2. Moreover, Reference 32 is not correct for the definition and description of TFCA.
3. Reference 44 (Yuan Y et al., 2021) is not correct for the description of extracellular vesicles (EVs).
4. Reference 45, 46 and 47 are not correct
5. I’m not satisfied by response to my concern 3. Although the authors replied to have added the suggested references I did not find them in the revised version!
6. References 52-54 have long non coding RNA as subject, but they are in text referred to microRNA!
7. Reference 57-59 are not appropriate for the sentence.
8. In the reference list lacks the reference Marsh et al. 2016.cited in the text at page 19 line 432.
9. Ref 145 is not completed but is not correct for the sentence.
10. Ref 238 might be deleted.
11. Ref 272 might be deleted.
Some sentences in the text are not clear and exhaustive.
1. Lines 61-62 page 5
2. Lines 131-134 page 5.
3. Line 141 page 6.
4. Lines 175-176 page 7 (genes are not controlled by miRNAs!)
5. lines 178-181 page 7
6. Lines 231-233 page 10.
7. lines 457-459 page 20 might be deleted.
8. lines 539-542 page 25
Lines 258-270 regards lncRNAs and are in the section 3.2 circRNAs and atherosclerosis!!!!!!
Table 5 lists drugs (canakinumab, succinobucol, torcetrapib, etc) used in clinical trials to stabilize plaque or attenuate atherosclerosis, but the authors did not describe which ncRNAS are the putative targets of these drugs. Consequently, I do not agree with the authors’ sentence “The results from these and other clinical trials (Table 5) suggest that targeting ncRNA to stabilize plaque or attenuate atherosclerosis will be of significance” (lines 553-555, page 25).
Author Response
Response to the Comments of Reviewer #3:
Comments: I noticed that the authors have made all the changes I outlined and modified the first 3 references; I think these new references are more appropriate. However, the authors must check accurately the reference list and the references cited in the text.
Response: Thank you for your comments. We respond to the individual comments below.
Concern 1. For example, the reference 13 (Zhu K et al., 2019) refers to diabetic diseases and ref 16 (Zhang F et al., 2018) to cardiovascular disease. However, in the text ref 13 is linked to CVD and 16 to diabetes!
Response: The references mentioned above are revised to the correct the order.
Concern 2. Moreover, Reference 32 is not correct for the definition and description of TFCA.
Response: The correct reference has been updated in the manuscript. (PMID 16631505)
Concern 3. Reference 44 (Yuan Y et al., 2021) is not correct for the description of extracellular vesicles (EVs).
Response: The appropriate references have been added in the manuscript.
Concern 4. Reference 45, 46 and 47 are not correct
Response: The references have been updated in the further revised manuscript.
Concern 5. I’m not satisfied by response to my concern 3. Although the authors replied to have added the suggested references I did not find them in the revised version!
Response: The suggested references have been added now in the further revised manuscript.
Concern 6. References 52-54 have long non coding RNA as subject, but they are in text referred to microRNA!
Response: The correct references have been included in the further revised manuscript.
Concern 7. Reference 57-59 are not appropriate for the sentence.
Response: The references 57-59 have been deleted in the further revised manuscript.
Concern 8. In the reference list lacks the reference Marsh et al. 2016.cited in the text at page 19 line 432.
Response: The reference has been updated in the reference list. (PMID 27233758)
Concern 9. Ref 145 is not completed but is not correct for the sentence.
Response: The ref 145 has been removed from the manuscript.
Concern 10. Ref 238 might be deleted.
Response: The ref 238 is deleted from the manuscript.
Concern 11. Ref 272 might be deleted.
Response: The ref 272 is also deleted from the manuscript.
Response to the Additional Comments:
Some sentences in the text are not clear and exhaustive.
Concern 1. Lines 61-62 page 5
Response: The lines 61-62 have been removed in the further revised manuscript.
Concern 2. Lines 131-134 page 5.
Response: The lines 131-134 have been removed from the manuscript.
Concern 3. Line 141 page 6.
Response: The line 141 has been deleted in the further revised manuscript.
Concern 4. Lines 175-176 page 7 (genes are not controlled by miRNAs!)
Response: The lines 175-176 have been corrected in the revised MS.
Concern 5. lines 178-181 page 7
Response: The lines 178-181 are corrected in the MS.
Concern 6. Lines 231-233 page 10.
Response: The lines 231-233 have been revised in the manuscript.
Concern 7. lines 457-459 page 20 might be deleted.
Response: The lines are removed in the revised manuscript.
Concern 8. lines 539-542 page 25
Response: The lines 539-542 have been corrected in the further revised manuscript.
Concern 9. Lines 258-270 regards lncRNAs and are in the section 3.2 circRNAs and atherosclerosis!!!!!!
Response: The lines 258-270 have been rearranged into the correct order in the further revised manuscript.
Concern 10. Table 5 lists drugs (canakinumab, succinobucol, torcetrapib, etc) used in clinical trials to stabilize plaque or attenuate atherosclerosis, but the authors did not describe which ncRNAS are the putative targets of these drugs. Consequently, I do not agree with the authors’ sentence “The results from these and other clinical trials (Table 5) suggest that targeting ncRNA to stabilize plaque or attenuate atherosclerosis will be of significance” (lines 553-555, page 25).
Response: The research studies investigating this correlation are limited in the literature. The available studies and clinical trials (NCT03603431, NCT03494712, NCT02603224, and NCT04045405) have discussed the role of miR-92a, miR-29b, and miR-132 in association with cutaneous healing and cardiac fibrosis, both having similar pathogenesis of inflammation and ECM remodelling.
The authors sincerely thank the reviewers for their constructive criticisms. We hope that the further revised manuscript is now suitable for publication in IJMS.